# Design of Species-Specific PCR Primers That Target the *aac(6′)-Ii* Gene for the Rapid Detection of *Enterococcus faecium*

**Slavil Peykov [1,2,3], Tanya Strateva [3] and Svetoslav Dimov [1,*]**

1   Department of Genetics, Faculty of Biology, Sofia University, St. Kliment Ohridski, 1164 Sofia, Bulgaria
2   BioInfoTech Laboratory, Sofia Tech Park, 1784 Sofia, Bulgaria
3   Department of Medical Microbiology, Faculty of Medicine, Medical University of Sofia, 1431 Sofia, Bulgaria
*   Correspondence: svetoslav@biofac.uni-sofia.bg

**Abstract:** In this study, we used the sequence of the *aac(6′)-Ii* gene, which is responsible for the intrinsic low-level aminoglycoside resistance of *Enterococcus faecium*, to design novel species-specific primers. Two oligonucleotide pairs named EF_200 and EF_120 were designed, generating amplification products with sizes of 200 bp and 120 bp, respectively. They were successfully applied for the identification of various isolates of clinical or environmental origins in both pure cultures and complex food samples. The obtained results indicated that both primer pairs permitted the highly specific, simple, fast and inexpensive detection of *E. faecium* isolates.

**Keywords:** *Enterococcus faecium*; ESKAPE; species-specific PCR; *aac(6′)-Ii*

## 1. Introduction

Enterococci were first discovered in human fecal flora in 1899. They include facultative anaerobic Gram-positive cocci that are common inhabitants of the gastrointestinal tracts of humans and almost all land animals [1]. Today, their group consists of more than 50 different species, many of them with ambivalent roles in regard to the health status of the inhabited macro-organisms [2]. The *Enterococcus faecium* isolates are a typical example of enterococci with such a dualistic nature. Many of them have been identified as part of the normal microbiota of various foods. Certain isolates even possess potential probiotic properties, as well as the so-called "generally recognized as safe (GRAS)" status [3]. On the other hand, *E. faecium* is also a known member of the ESKAPE (*E. faecium*, *Staphylococcus aureus*, *Klebsiella pneumoniae*, *Acinetobacter baumannii*, *Pseudomonas aeruginosa* and *Enterobacter* species) group of pathogens that represent a substantial therapeutic problem worldwide [4]. This group causes serious and life-threatening infections in humans, including urinary tract infections, blood stream infections and endocarditis. Moreover, the *E. faecium* isolates have an intrinsic resistance to some widely used antibiotics (e.g., penicillin, ampicillin and most cephalosporins) and can easily acquire glycopeptide resistance as well [5]. All these findings highlight this species as a leading cause of serious hospital-acquired infections.

Due to their unique dualistic nature, *E. faecium* strains still remain an unsolved puzzle in terms of their recognition as useful or harmful microorganisms. Their investigation has attracted a great deal of attention in numerous research fields, such as clinical microbiology, environmental metagenomics, the microbiological analysis of foods and food processing environments, etc. This requires the development of an accurate, easy and cost-efficient procedure for the identification of *E. faecium* in samples of various origins. The phenotypic and biochemical similarities of enterococci have make species identification by conventional tests a challenging, complex, time-consuming and expensive task [6]. To overcome these issues, most of the recently developed strategies for *E. faecium* identification are based on molecular techniques. Since the substitution of the initially implemented thermolabile Klenow fragment with a thermostable polymerase from *Thermus aquaticus*, the polymerase chain reaction has been proven as a well-established method for the detection

and typing of different microbial agents in clinical and/or environmental samples [7]. In the study presented here, we describe a PCR-based identification procedure for *E. faecium* detection. It utilizes newly designed species-specific primers targeting the *aac(6′)-Ii* gene, which is responsible for the low-level intrinsic aminoglycoside resistance observed in these microorganisms.

## 2. Results

The *aac(6′)-Ii* gene was identified in 99.19% of the completely sequenced *E. faecium* genomes and plasmids and, at the same time, it was absent in the other enterococci [8]. Two primer pairs that target its sequence were designed and synthesized. Their details are given in Table 1.

**Table 1.** Primer pairs that target the *aac(6′)-Ii* gene.

| Primer Name | Sequence (5′-3′) | Amplicon Length (bp) | Ta (°C) |
|---|---|---|---|
| Ef_200_F<br>Ef_200_R | TAGTAGGATTTATTGGTGCAATCCC<br>TTGACTTAACGTTGTTCCATGGTC | 200 | 60 |
| Ef_120_F<br>Ef_120_R | ACGAAAGAACCAAATAGGTACTCG<br>TGACTTAACGTTGTTCCATGGTC | 120 | 60 |

Ta—temperature of annealing.

Both primer pairs successfully generated PCR products of the expected size when genomic DNA from the *E. faecium* EFD isolate [9] was used as a template. None of them produced bands when the amplification reactions were supplemented with genomic DNA from the vancomycin-resistant *Enterococcus faecalis* BG475 [10]. These results are shown in Figure 1a.

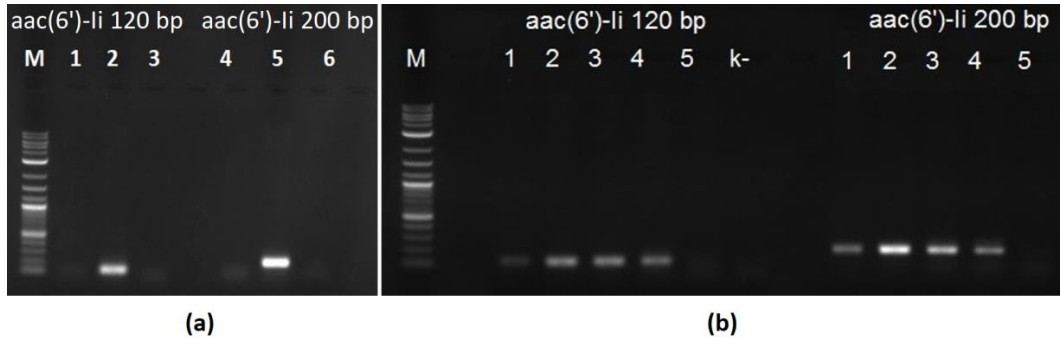

**Figure 1.** PCR amplifications with the primer pairs Ef_200 and Ef_120. (**a**) M—Quick-Load Purple 1 kb DNA Ladder (NEB); 1—gDNA from *E. faecalis* BG475; 2—gDNA from *E. faecium* EFD; 3—negative control; (**b**) M—Quick-Load Purple 1 kb DNA Ladder (NEB); 1–4—gDNA from *E. faecium* FM486, FM493, FM494 and FM496; 5—negative control.

Next, the genomes of four clinical vancomycin-resistant *E. faecium* isolates were sequenced and assembled. The resulting draft genome sequences were analyzed, and all of them contained the *aac(6′)-Ii* resistance determinant. Genomic DNA from these isolates was used as a PCR matrix with both the Ef_200 and Ef_120 primer pairs. All the amplification reactions yielded products of the expected sizes (Figure 1b).

All 71 publicly available chromosome-level sequenced genomes of the other enterococci found in the prokaryotic reference genomes database were used to create a panel of sequences that were then *in silico* analyzed with both primer pairs using the Primer-BLAST tool (all used genomes are given in the Supplementary Materials) [11].

The obtained results confirmed that our primer pairs always generated products of the expected sizes when we used *E. faecium* genomes as the templates. On the contrary,

when other genomes were utilized as the matrixes, no PCR products of the expected or similar sizes were predicted.

The multiple sequence alignment of all the primer-annealing regions identified in the 24 *E. faecium* genomes included in our panel indicated that these locations were highly conserved, and no SNPs were identified (Figure 2).

|  | EF200F | EF200R | EF120F | EF120R |
|---|---|---|---|---|
| NZ_LR607349.1 | TAGTAGGATTTATTGGTGCAATCCC | GACCATGGAACAACGTTAAGTCAA | ACGAAAGAACCAAATAGGTACTCG | GACCATGGAACAACGTTAAGTCA |
| NZ_LR607355.1 | TAGTAGGATTTATTGGTGCAATCCC | GACCATGGAACAACGTTAAGTCAA | ACGAAAGAACCAAATAGGTACTCG | GACCATGGAACAACGTTAAGTCA |
| NZ_LR607353.1 | TAGTAGGATTTATTGGTGCAATCCC | GACCATGGAACAACGTTAAGTCAA | ACGAAAGAACCAAATAGGTACTCG | GACCATGGAACAACGTTAAGTCA |
| NZ_CP012465.1 | TAGTAGGATTTATTGGTGCAATCCC | GACCATGGAACAACGTTAAGTCAA | ACGAAAGAACCAAATAGGTACTCG | GACCATGGAACAACGTTAAGTCA |
| NZ_CP012447.1 | TAGTAGGATTTATTGGTGCAATCCC | GACCATGGAACAACGTTAAGTCAA | ACGAAAGAACCAAATAGGTACTCG | GACCATGGAACAACGTTAAGTCA |
| NZ_CP012436.1 | TAGTAGGATTTATTGGTGCAATCCC | GACCATGGAACAACGTTAAGTCAA | ACGAAAGAACCAAATAGGTACTCG | GACCATGGAACAACGTTAAGTCA |
| NZ_CP064432.1 | TAGTAGGATTTATTGGTGCAATCCC | GACCATGGAACAACGTTAAGTCAA | ACGAAAGAACCAAATAGGTACTCG | GACCATGGAACAACGTTAAGTCA |
| NZ_CP060857.1 | TAGTAGGATTTATTGGTGCAATCCC | GACCATGGAACAACGTTAAGTCAA | ACGAAAGAACCAAATAGGTACTCG | GACCATGGAACAACGTTAAGTCA |
| NZ_CP012471.1 | TAGTAGGATTTATTGGTGCAATCCC | GACCATGGAACAACGTTAAGTCAA | ACGAAAGAACCAAATAGGTACTCG | GACCATGGAACAACGTTAAGTCA |
| NZ_CP012440.1 | TAGTAGGATTTATTGGTGCAATCCC | GACCATGGAACAACGTTAAGTCAA | ACGAAAGAACCAAATAGGTACTCG | GACCATGGAACAACGTTAAGTCA |
| NZ_CP064443.1 | TAGTAGGATTTATTGGTGCAATCCC | GACCATGGAACAACGTTAAGTCAA | ACGAAAGAACCAAATAGGTACTCG | GACCATGGAACAACGTTAAGTCA |
| NZ_CP066730.1 | TAGTAGGATTTATTGGTGCAATCCC | GACCATGGAACAACGTTAAGTCAA | ACGAAAGAACCAAATAGGTACTCG | GACCATGGAACAACGTTAAGTCA |
| NZ_CP064417.1 | TAGTAGGATTTATTGGTGCAATCCC | GACCATGGAACAACGTTAAGTCAA | ACGAAAGAACCAAATAGGTACTCG | GACCATGGAACAACGTTAAGTCA |
| NZ_CP012454.1 | TAGTAGGATTTATTGGTGCAATCCC | GACCATGGAACAACGTTAAGTCAA | ACGAAAGAACCAAATAGGTACTCG | GACCATGGAACAACGTTAAGTCA |
| NZ_CP036151.1 | TAGTAGGATTTATTGGTGCAATCCC | GACCATGGAACAACGTTAAGTCAA | ACGAAAGAACCAAATAGGTACTCG | GACCATGGAACAACGTTAAGTCA |
| NZ_CP025392.1 | TAGTAGGATTTATTGGTGCAATCCC | GACCATGGAACAACGTTAAGTCAA | ACGAAAGAACCAAATAGGTACTCG | GACCATGGAACAACGTTAAGTCA |
| NZ_CP025389.1 | TAGTAGGATTTATTGGTGCAATCCC | GACCATGGAACAACGTTAAGTCAA | ACGAAAGAACCAAATAGGTACTCG | GACCATGGAACAACGTTAAGTCA |
| NZ_CP043865.1 | TAGTAGGATTTATTGGTGCAATCCC | GACCATGGAACAACGTTAAGTCAA | ACGAAAGAACCAAATAGGTACTCG | GACCATGGAACAACGTTAAGTCA |
| NZ_CP092583.1 | TAGTAGGATTTATTGGTGCAATCCC | GACCATGGAACAACGTTAAGTCAA | ACGAAAGAACCAAATAGGTACTCG | GACCATGGAACAACGTTAAGTCA |
| NZ_LR536658.1 | TAGTAGGATTTATTGGTGCAATCCC | GACCATGGAACAACGTTAAGTCAA | ACGAAAGAACCAAATAGGTACTCG | GACCATGGAACAACGTTAAGTCA |
| NZ_CP101669.1 | TAGTAGGATTTATTGGTGCAATCCC | GACCATGGAACAACGTTAAGTCAA | ACGAAAGAACCAAATAGGTACTCG | GACCATGGAACAACGTTAAGTCA |
| NZ_CP084886.1 | TAGTAGGATTTATTGGTGCAATCCC | GACCATGGAACAACGTTAAGTCAA | ACGAAAGAACCAAATAGGTACTCG | GACCATGGAACAACGTTAAGTCA |
| NZ_LR607382.1 | TAGTAGGATTTATTGGTGCAATCCC | GACCATGGAACAACGTTAAGTCAA | ACGAAAGAACCAAATAGGTACTCG | GACCATGGAACAACGTTAAGTCA |
| NZ_CP025022.1 | TAGTAGGATTTATTGGTGCAATCCC | GACCATGGAACAACGTTAAGTCAA | ACGAAAGAACCAAATAGGTACTCG | GACCATGGAACAACGTTAAGTCA |
|  | ************************* | ************************ | ************************ | *********************** |

**Figure 2.** Corresponding regions of the primer pairs Ef_200 and Ef_120 in the publicly available chromosome-level sequenced *E. faecium* genomes. The multiple sequence alignment was generated with Clustal Omega. No SNPs were found. * - position which have a single, fully conserved residue.

Later, we tested both primer pairs with the gDNA extracted from other enterococci, including one strain of *Enterococcus hirae* and one strain of *Enterococcus avium*. The results are shown on Figure 3.

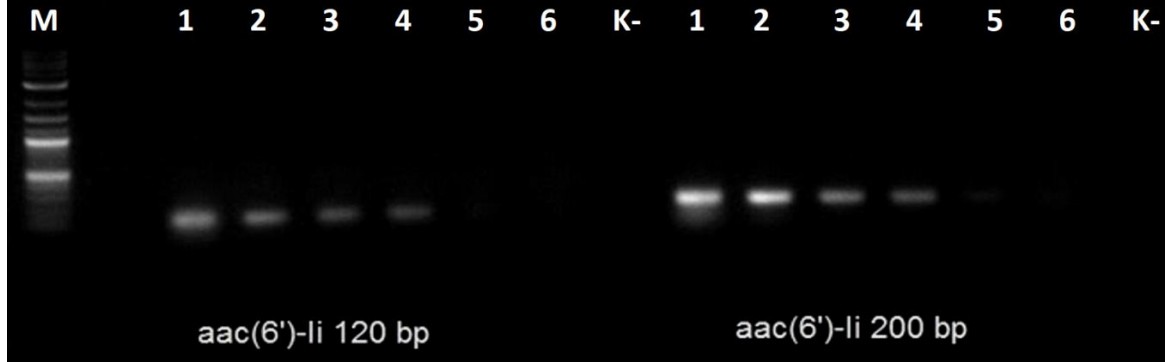

**Figure 3.** Validation of the pairs Ef_200 and Ef_120. M—Quick-Load Purple 1 kb DNA Ladder (NEB); 1–4—gDNA from *E. faecium* FM486, FM493, FM494 and FM496; 5—gDNA from *E. hirae* 8326; 6—gDNA from *E. avium* 983; K⁻—negative control.

In parallel with the PCR tests, all four clinical *E. faecium* isolates were observed to exhibit an aminoglycoside resistance phenotype (Figure 4). This finding suggested that the presence of *aac(6′)-Ii* can be detected independently, thus providing an additional method for cross-verifying the results of the proposed PCR-based identification procedure.

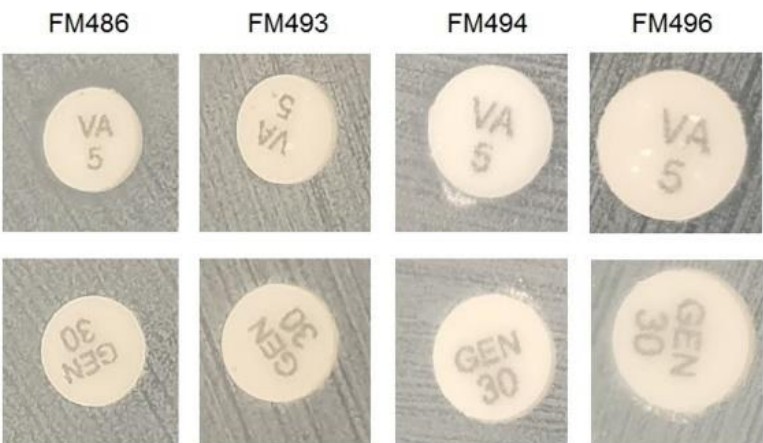

**Figure 4.** Phenotypical analysis of the glycopeptide and aminoglycoside resistance of all four clinical *E. faecium* isolates. The disk diffusion method was used to test the susceptibility of the strains to vancomycin and gentamicin.

In order to determine the lower detection limit of our PCR-based assay, we used serial dilutions of plasmid vectors that contained the corresponding products generated by the described primer pairs. The lower detection limits in both cases were determined to be in the range of 10–100 copies of the genome equivalent per reaction.

Finally, the Ef_120 primer pair enabled the successful detection of *E. faecium* in commercially available Bulgarian yogurt that was initially contaminated with the FM493 clinical isolate (Figure 5).

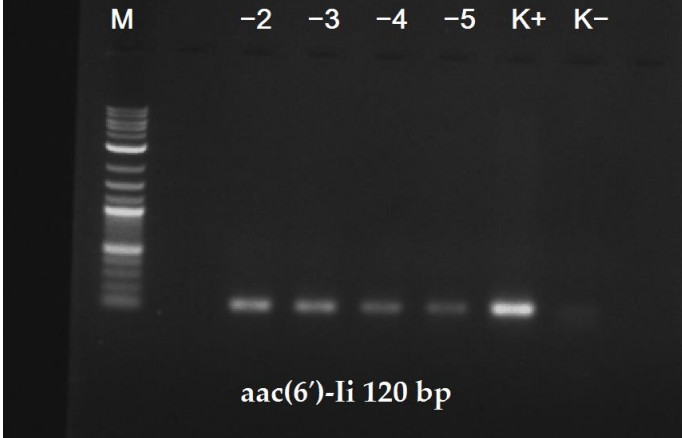

**Figure 5.** The application of the primer pair Ef_120 in the detection of *E. faecium* in yogurt. M—Quick-Load Purple 1 kb DNA Ladder (NEB); −2 to −5—total DNA from yogurt samples with decreasing numbers of FM493 cells; K+—gDNA from pure culture *E. faecium* FM493; K−—negative control.

This result suggested that the identification strategy presented herein can be applied not only for the detection of this species in pure cultures, but also in food control procedures using complex samples, without additional cultivation and/or enrichment steps.

## 3. Discussion

Various non-PCR-based methods for the detection of *E. faecium* are currently in use, including conventional biochemical tests [12], automated systems [13], matrix-assisted laser desorption ionization time-of-flight mass spectrometry analysis [14], etc. These techniques are affected by some serious limitations, such as their low time/cost efficiency, significant number of misidentified isolates and requirements for expensive equipment with complex workflows. In order to compensate for these disadvantages, many researchers

have developed PCR methodologies that enable the species-specific detection of *E. faecium* isolates in different clinical and environmental samples [15]. It should be noted that such PCR-based approaches also possess some restrictions. Probably, the most important of these are the higher rate of false positive results due to the nonspecific annealing of the primers and the possibility of a false negative misidentification of a given isolate due to sequence variations in the primer-corresponding sequences. The easiest way to compensate for these limitations is to use carefully designed primer pairs for such assays. Currently, this can be achieved far more easily than ever before, because the available genomic databases include tens of thousands of bacterial genomes that enable the identification of species-specific sequences and their highly conserved regions. Moreover, recent developments in the gene synthesis field have enabled the cost-efficient production of such genome fragments that can be used for the validation of the primers' specificity.

Most PCR-based methods for *E. faecium* detection utilize the primer pair designed by Cheng et al. [6], which produces a 658-base-pair product upon amplification. It was designed by subtractive hybridization, and initially the target DNA region was not known.

The strategy that is described in the current work utilizes a different approach. It relies on the identification of a species-specific sequence in the core genome of *E. faecium*. Once this DNA region is detected, the interested researcher can design various primer pairs with flexible parameters. Literature mining revealed that the low-level aminoglycoside resistance provided by *aac(6′)-Ii* is ubiquitously present among the *E. faecium* isolates, making this genetic determinant the perfect candidate for our work. It was further confirmed to be species-specific based on the information found in the comprehensive antibiotic resistance database (CARD) [8]. A retrospective literature analysis of the *aac(6′)-Ii* gene, after its selection, revealed that it had already been used as an *E. faecium*-specific probe in Southern blot analyses previously [16]. It is not known why the designing of species-specific PCR primers based on this gene has not been attempted until now.

In addition to all other benefits of a rational primer design, the *aac(6′)-Ii* coding sequence possesses one other advantage: its presence can be detected by the exhibited aminoglycoside resistance phenotype. As demonstrated, a simple disc diffusion method can be used for this purpose, enabling the independent phenotypical validation of the PCR amplification. Nevertheless, it is worth mentioning that the aminoglycoside resistance identified in the candidate isolates may be caused by other mechanisms, especially in the case of clinical samples. This suggests that the phenotype alone is by no means species-specific and can be used only in addition to the PCR amplification.

The lower detection limit of the method proposed herein is higher compared to other 16S-based assays for bacterial taxonomy. This is due to the fact that the locus selected by us is a single copy region in the enterococcal genome. It is worth mentioning this disadvantage of PCR-based detection that uses unique genomic regions: it usually requires more genomic equivalents per reaction for a reliable detection.

In conclusion, it should be mentioned that the *aac(6′)-Ii* sequence was successfully used for the design of two novel species-specific primer pairs. They permitted the highly specific, simple, fast and inexpensive detection of *E. faecium* isolates in pure cultures and complex food samples. Moreover, the small size of the product generated by pair Ef_120 could potentially enable its further application in quantitative PCR analyses.

## 4. Materials and Methods

### 4.1. Primer Design

The *aac(6′)-Ii* sequence was obtained from GenBank (AAB63533.1) and was used for the blastn search on NCBI. The top 1000 hits were downloaded, and the unique sequences among them were used for the primer design with the Primer-BLAST tool [11]. The option for designing primers common to a group of sequences was applied. The generated primer pairs were synthesized by Sigma-Aldrich (Darmstadt, Germany). All sequences used for the generation of the primer pairs, as well as the corresponding multiple sequence alignments produced by Primer-BLAST, are available in the Supplementary Materials.

### 4.2. Isolates Used

The isolates *E. faecium* EFD and *E. faecalis* BG475 have been described previously [9,10]. Their corresponding genome sequences are available at GenBank with the corresponding access numbers JAANIF010000000 and JAHZNN010000000. The four vancomycin-resistant *E. faecium* strains of clinical origin, FM486, FM493, FM494 and FM496, were obtained from the collection of the Medical University of Sofia. The *E. hirae* 8326 and *E. avium* 983 strains were obtained from the National Bank of Industrial Microorganisms and Cell Cultures, located in Sofia, Bulgaria (accessed on 15 September 2011).

### 4.3. Genomic DNA Extraction

Genomic DNA was extracted from all the isolates using a DNeasy Blood & Tissue Kit (QIAGEN, Hilden, Germany), following the provided protocol for Gram-positive bacteria without modification. A total of 0.5 µL of the eluted DNA was used as the PCR matrix.

### 4.4. PCR Amplifications

All PCR amplifications were performed in a volume of 20 µL using a 2×Horse-Power Green-Taq DNA Polymerase Master Mix (Canvax, Cordoba, Spain) according to the provided protocol. The amplification program comprised of 30 cycles with the following conditions: 94 °C 5 min, 30× (94 °C 35 s, 60 °C 35 s, 72 °C 30 s), 72 °C 2 min and 4 °C ∞.

### 4.5. Species Identification of the Isolates

The species identification of the isolates was conducted by analyzing the corresponding assembled draft genome sequences using the Microbial Genomes Atlas (MiGA) web server [17]. The included workflow of the NCBI Genome Database, for the prokaryotic section, was followed with the default settings.

### 4.6. Whole-Genome Sequencing

The whole-genome sequencing of isolates FM486, FM493, FM494 and FM496 was performed using DNA nanoball sequencing technology, as previously described [18]. In brief, the extracted genomic DNA was randomly size-fragmented using a Covaris g-TUBE device, and the fragments were size-selected using magnetic beads to obtain an average size of 200 to 400 bp. The purified fragments were end-repaired, 3′-adenylated, ligated to adapters and then PCR amplified. The final generated library had an insert size of 350 bp, and it was loaded onto an MGISEQ-2000 platform (BGI Group, Hong Kong, China). There, the sequencing step was performed, generating 2 × 150 bp paired-end reads.

### 4.7. Draft Genome Assembly

All the steps of the quality control, raw reads preprocessing and draft genome assembly were carried out on the Galaxy online platform [19]. Default parameters were used for all the following software tools unless otherwise specified. The entire procedure was performed as previously described [18], using the following tools: FastQC v0.11.9 [20], Trimmomatic v0.38 [21] and SPAdes v3.12.0 [22]. The draft genome sequences of the isolates *E. faecium* EFD and *E. faecalis* BG475 were obtained beforehand [9,10].

### 4.8. Antimicrobial Susceptibility Testing

The antimicrobial susceptibility of the implemented *E. faecium* isolates, which were of clinical origin, to the antimicrobial agents vancomycin and gentamicin was determined by the disc diffusion method, according to the European Committee on Antimicrobial Susceptibility Testing (EUCAST) recommendations.

### 4.9. Determination of the Lower Detection Limit

In order to determine the lower detection limit of our PCR-based assay, both products generated by our primer pairs were cloned in the pSDTV vector via TA cloning. Plasmid DNA was isolated, quantified and diluted in double-distilled water. Using serial dilu-

tions, the lower detection limit was determined by running standard PCR amplifications in triplicate.

*4.10. Detection of E. faecium in the Yogurt Samples*

The culture of *E. faecium* FM493 was prepared in liquid BHI medium and cultivated overnight at 37 °C. The next morning, the yogurt aliquots were contaminated with 50 μL of decreasing serial decimal dilutions. Next, the total DNA was extracted using a DNeasy Power Food Microbial Kit (QIAGEN, Hilden, Germany), following the provided protocol. A total of 2 μL of the eluted DNA was used as the PCR matrix.

**Supplementary Materials:** The following are available online at https://www.mdpi.com/article/10.3390/bacteria1030014/s1, File S1: Used sequences for primer design, File S2: Primer-BLAST MSA, File S3: Genomes used for in silico validation.

**Author Contributions:** Conceptualization, S.P.; methodology, S.P. and S.D.; software, S.P.; validation, S.P., T.S. and S.D.; formal analysis, S.P., T.S. and S.D.; investigation, S.P.; resources, S.P., T.S. and S.D.; data curation, S.P., T.S. and S.D.; writing—original draft preparation, S.P.; writing—review and editing, S.P., T.S. and S.D.; visualization, S.P., T.S. and S.D.; supervision, T.S. and S.D.; project administration, S.D.; funding acquisition, S.D. All authors have read and agreed to the published version of the manuscript.

**Funding:** This study was funded by the Bulgarian National Research Fund under grant no. КП-06-Н26/8 from 17.12.2018.

**Institutional Review Board Statement:** Not applicable.

**Informed Consent Statement:** Not applicable.

**Data Availability Statement:** Not applicable.

**Acknowledgments:** We gratefully thank M. Dimitrova for the technical support.

**Conflicts of Interest:** The authors declare no conflict of interest.

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
