# Peer review of "Design of Species-Specific PCR Primers That Target the aac(6′)-Ii Gene for the Rapid Detection of Enterococcus faecium"

_2674-1334, doi:10.3390/bacteria1030014_

Round 1

Reviewer 1 Report

Dear Authors,

The article is well design and planned, in the introduction you briefly described the most relevant information and posted the aim of the study.  Results and discussion are clearly presented. Materials and methods are well developed and thought out. 

Please find the revised version in the attachment.

Author Response

Dear Sir/Madam Reviewer,

We would like to express our gratitude to you for reviewing our manuscript entitled “Design of species-specific PCR primers that target the aac(6')-Ii gene for rapid detection of Enterococcus faecium

We did our best to take into consideration all that was suggested by you and the reviewers, in order to improve our manuscript and meet the high standards for publishing articles in Bacteria.

The corrections made are highlighted in yellow in the text. We included more detailed validation of our primer pairs with an additional figure, all genomes used are given with access numbers and the lower detection limit was determined as suggested by the reviewers.

We sincerely apologize for the delay but the experimental part of the revision took longer than initially expected.

Reviewer 2 Report

In this study, the authors used the sequence of the aac(6')-Ii gene responsible for the intrinsic low level aminoglycoside resistance in E. faecium to design novel species-specific primers. The results indicated that both primer pairs permitted highly specific, simple, fast and inexpensive 16 detection of E. faecium isolates. The study design and presentation are detailed. The data presented are generally strong, and appear convincing. It would benefit with further experiments to help strengthen the main conclusions and to better understand.

Also, it would be perfect if the authors could detected the lower detection limit.

I just want to point out that maybe the authors can also emphasize the disadvantages of this method in the discussion part by comparing other methods.

Some language errors should be corrected, such as some name of the bacteria should be italic.

Author Response

Dear Sir/Madam Reviewer,

We would like to express our gratitude to you for reviewing our manuscript entitled “Design of species-specific PCR primers that target the aac(6')-Ii gene for rapid detection of Enterococcus faecium

We did our best to take into consideration all that was suggested by you and the reviewers, in order to improve our manuscript and meet the high standards for publishing articles in Bacteria.

The corrections made are highlighted in yellow in the text. We included more detailed validation of our primer pairs with an additional figure, all genomes used are given with access numbers and the lower detection limit was determined as suggested by the reviewers.

We sincerely apologize for the delay but the experimental part of the revision took longer than initially expected

Reviewer 3 Report

In this study, Peykov et al., have designed two nucleotide primer pairs to amplify the region of aac(6')-li gene that can be used for detection of Enterococcus faecium. The study is very technical molecular biology study with very limited novelty. I have following comments:

a) The scope for this article is very limited. The experimental data is limited to just PCR reactions with two primer pairs. Therefore, I have failed to understand if it is a short communication or a full-fledged research article.

b) PCR tests are although rapid compared to biochemical tests but are error prone leading to false positive results. Therefore, are not reliable. qRT-PCR are more sensitive and accurate. So, I do not see wide application of this method in lab setting for clinical specimens analysis. 

c) The entire study is designed around designing two primers against a specific gene that is present in Enterococcus faecium. So overall, I do not see any novelty in this study. It is well known that primers can be designed to amplify a specific genomic region. It can be performed against any bacteria/pathogen in the same way.

Author Response

(The authors gave the same response as above.)

Round 2

Reviewer 3 Report

I do not see much improvement in the manuscript therefore my previous review stands as it is. Moreover, the author has provided a very general response in rebuttal letter without referring to any of my comments mentioned in my previous review. In absence of an appropriate rebuttal letter, it is difficult for me to confirm if authors has considered any of my comment to make necessary amendments. 

Author Response

                Dear Sir/Madam Reviewer,

We would like to express our gratitude to you for the second round of revision of our manuscript entitled “Design of species-specific PCR primers that target the aac(6')-Ii gene for rapid detection of Enterococcus faecium”.

We did our best to take into consideration all that was suggested by you in order to improve the quality our manuscript and meet the high standards for publishing articles in Bacteria.

Below we present our point-by-point answers to your comments.

Reviewer:

Comment 1: The scope for this article is very limited. The experimental data is limited to just PCR reactions with two primer pairs. Therefore, I have failed to understand if it is a short communication or a full-fledged research article.

This work in its current form is presented as a research article. We agree to a certain extent with the reviewer that it can be presented in other forms but the editor approved the title and the format before the first submission. Moreover, in our opinion, five figures and one table are too much for a short communication.

Comment 2: PCR tests are although rapid compared to biochemical tests but are error prone leading to false positive results. Therefore, are not reliable. qRT-PCR are more sensitive and accurate. So, I do not see wide application of this method in lab setting for clinical specimens analysis.

We completely agree with this remark of the reviewer. Indeed, the application of the proposed here method for clinical specimens analysis will be very limited in developed countries. Unfortunately, large part of the population worldwide lives in regions where MALDI-TOF MS devices, qPCR cyclers, and other modern equipment are not available and/or the cost of the required reagents for such analyses are way to high for the local healthcare systems. In these environments, the classical PCR still has vast applications as a quick and cost-efficient (both in terms off equipment and reagents) approach for clinical specimens analysis. Having this in mind, our method can contribute to the efforts for limiting the discrimination based on economic factors in this field.

Comment 3: The entire study is designed around designing two primers against a specific gene that is present in Enterococcus faecium. So overall, I do not see any novelty in this study. It is well known that primers can be designed to amplify a specific genomic region. It can be performed against any bacteria/pathogen in the same way.

The novelty, in our opinion, is that we present two new primer pairs for an important ESKAPE pathogen. They were extensively validated and can be readily used by anyone who is interested in E. faecium detection by PCR. Both primer pairs generate significantly shorter products compared to the other alternatives found in the literature. This allows reduction of the overall reaction time. Moreover, the presence of the targeted gene can be easily detected by its phenotype via disc diffusion method. According to our knowledge, none of the other primer pairs designed so far for E. faecium detection allows similar cross-validation.